# A Cross-Layer Optimization QoS Scheme in Wireless Multimedia Sensor Networks

**Shu Fan**

Audio-Visual and Image Technology Department, Criminal Investigation Police University of China, Shenyang 110854, China; fanshu@npuc.edu.cn; Tel.: +86-13940398360

**Abstract:** There are two main challenges in wireless multimedia sensors networks: energy constraints and providing DiffServ. In this paper, a joint flow control, routing, scheduling, and power control scheme based on a Lyapunov optimization framework is proposed to increase network lifetime and scheduling fairness. For an adaptive distribution of transmission opportunities, a differentiated queueing services (DQS) scheme is adopted for maintaining data queues. In the Lyapunov function, different types of queues are normalized for a unified dimension. To prolong network lifetime, control coefficients are designed according to the characteristics of the wireless sensor networks. The power control problem is proved to be a convex optimization problem and two optimal algorithms are discussed. Simulation results show that, compared with existing schemes, the proposed scheme can achieve a better trade-off between QoS performances and network lifetime. The simulation results also show that the scheme utilizing the distributed media access control scheme in scheduling performs best in the transmission of real-time services.

**Keywords:** Lyapunov optimization; wireless multimedia sensor networks; cross-layer control; QoS

## 1. Introduction

Real-time applications, including multimedia applications, have been developed as important services in various types of networks [1,2]. In wireless multimedia sensor networks, real-time and non-real-time applications coexist. In wireless multimedia sensor networks, there are two essential problems. Firstly, as real-time and non-real-time applications have different QoS requirements, it is valuable to design QoS schemes which provide DiffServ. Secondly, as wireless sensor networks are energy-constrained, it is desirable to prolong network lifetime and reduce energy consumption while satisfying the QoS demands of the applications.

The mainstream approach of QoS schemes providing DiffServ is to maintain different data queues for different applications. Queues are assigned different transmission priorities according to the types of applications that the queues support. The existing QoS schemes providing DiffServ can be divided into four categories of algorithms: media access control schemes [3,4], scheduling schemes [5–11], routing schemes [12–14], and cross-layer control schemes [15–17]. However, in these works, after initial allocation, the priorities of the queues cannot be re-adjusted according to the real-time end-to-end situation. In the case of a shortage of resources, services with low priorities might obtain hardly any transmission opportunities. In [18], a DQS (Differentiated Queueing Services) scheme was proposed in which packets of all services are buffered in the same data queue at each node. The queueing sequence of a packet in the queue is decided according to its remaining lifetime. This queue structure means that any packet can gain transmission opportunities according to its end-to-end delay requirement and end-to-end situation. In [19], a DQS scheme is applied to wireless mesh networks. In [20], a DQS-based queueing algorithm was proposed in which event QoS parameters

are used to reflect the actual need of non-end-to-end applications in wireless sensor networks. However, in [18–20] link scheduling was not considered.

Algorithms of energy consumption control consist of three categories: cluster-based schemes [21–28], optimization-based schemes [29–37], and backpressure-based schemes [38–43]. Cluster-based schemes can balance the energy consumption of nodes and prolong network lifetime through selecting cluster-head nodes and forming clusters according to some ingenious criteria. TDMA (Time Division Multiple Access) scheduling is mostly utilized in clusters. However, cluster-based schemes may result in low utilization of wireless resources when nodes are non-uniformly distributed. This may degrade QoS performances. In optimization-based schemes, network utility, network lifetime, or energy consumption are used as the optimization objective. Optimization-based QoS schemes are designed based on the process of solving optimization problems. Unlike optimization-based schemes, backpressure optimization problems are established based on a Lyapunov optimization framework with time being slotted, and the lower bounds of the time average values of optimization utilities achieved by backpressure-based schemes can be pushed arbitrarily to the optimal values [44]. In [38–40], bias factors relating to the residual energy of nodes or distances between nodes are introduced into link-weight calculation to control transmission scheduling. In [41] and [42], both data queues and energy queues are utilized in Lyapunov functions for cross-layer control. In [43], the shortest path is combined with a backpressure scheme in the process of next-hop node selecting to prolong network lifetime. However, two problems are ignored in existing optimization-based and backpressure-based schemes. Firstly, the characteristic that the lifetimes of nodes in the area surrounding the sink node determine the network lifetime is not considered, which causes the trade-off between data transmission and the energy consumption of nodes in the area surrounding the sink node to be ignored. Secondly, how to set weights of data queues and energy queues in the optimization framework is not taken account. The above two problems may cause a reduction in the network lifetime.

Thus, existing QoS schemes for wireless multimedia sensor networks do not achieve a good trade-off between QoS performances and network lifetime. In addition, the scheduling schemes for data packets of different types of applications also need improvement. To solve the above problems, this paper proposes (based on a Lyapunov optimization framework) a joint flow control, routing, scheduling, and power control algorithm in wireless multimedia sensor networks. The key contributions of this paper can be summarized as follows:

- Considering the characteristics of wireless sensor networks, distance coefficients are introduced into the Lyapunov function to increase network lifetime while maintaining QoS performances.
- To unify the dimension, different types of queues are normalized in the Lyapunov optimization framework.
- A DQS scheme and delay coefficients are utilized to adaptively allocate scheduling priorities to packets belonging to different services.

The remainder of this paper is organized as follows. Section 2 introduces the system model and queue structures. In Section 3, the algorithm is designed based on Lyapunov optimization. The simulation results are given in Section 4. Finally, the conclusions are provided in Section 5.

## 2. Network Model and Design of Queues

### 2.1. Network Model

The wireless multimedia sensor network consists of common nodes and multimedia nodes. Packets of non-real-time sessions are generated from common nodes. Multimedia nodes are the source nodes of real-time sessions. The destination node of all packets in the network is the sink node, which is denoted by $d^*$. $m$ denotes a non-real-time or real-time unicast session that has one source-destination pair. $s_m$ is the source node of session $m$. $M$ represents the set of sessions. Obviously, each node is the source node of one session. Packets from the source nodes traverse one or multiple wireless hops before arriving at the sink node. Both common nodes and multimedia nodes can relay

data packets. Multimedia nodes are initialized with higher energy than common nodes. The sink node is assumed to have enough energy in the running process. $\bar{x}$ denotes the time average value of x(t), which is a generic variable. $\bar{x}$ is calculated using $\bar{x} = \lim_{t \to \infty} \frac{1}{t} \sum_{\tau=0}^{t-1} E(x(\tau))$. The network runs in a time-slotted fashion. There are two channels in the networks, the control channel and the data channel. The radio frequencies used by the two channels are different. Nodes exchange control information on the control channel. When a node broadcasts local information on the control channel, other nodes can gain the information through monitoring the control channel. Data packets are transmitted on the data channel. If the link (n,j) that is between node n and j is used for data packets transmission in time slot t, we set $\alpha_{nj}(t) = 1$. If the link (n,j) is idle in time slot t, $\alpha_{nj}(t) = 0$. In this model, link scheduling is subjected to the following constraint:

$$\sum_{j \in O(n)} \alpha_{nj}(t) + \sum_{i \in I(n)} \alpha_{in}(t) \leq 1 . \tag{1}$$

$O(n)$ represents the set of nodes that can receive packets from node n. $I(n)$ denotes the set of nodes which can send packets to node n. Constraint (1) implies that each node can either transmit or receive data in the same time slot.

Assume that node n is the sending node and node i is the receiving node of the link (n,i). The transmission power of node n is subjected to the following constraint:

$$0 \leq p_{ni}^T \leq p_{n,\max}^T , \tag{2}$$

where $p_{n,\max}^T$ is the maximum transmission power that node n can support. The SINR (Signal to Interference plus Noise Ratio) at the receiving node i can be calculated as follows:

$$SINR_i = \frac{G_{ni} p_{ni}^T K_{ni}}{\sum_{k \neq n, k \neq i} G_{ki} p_{kk'}^T + n_i} , \tag{3}$$

where $k'$ is the target node of the transmission from sending node k. $K_{ni}$ denotes the processing gain of the CDMA system. $G_{ni}$ is the transmission loss from node n to node i. In this paper, $G_{ni} = \frac{1}{d_{ni}^4}$, where $d_{ni}$ is the distance between node n and node i. $n_i$ represents the noise at node i. The transmission capacity of link (n,i) can be derived as:

$$C_{ni} = B \cdot \log(1 + SINR_i) , \tag{4}$$

where $B$ denotes the bandwidth of data channel. If $SINR_i$ is high enough, $C_{ni}$ can also be calculated as follows:

$$C_{ni} \approx B \cdot \log(SINR_i) . \tag{5}$$

In the wireless networks, a link may be shared by several sessions. In the same time slot, the total data amount of all sessions transmitted on one link is constrained by the transmission data amount the link can support in a time slot. The constraint is as follows:

$$\sum_{m \in S(n,i)} \mu_{ni}^{(m)}(t) \leq C_{ni}(t) \cdot t_{slot} , \tag{6}$$

where $S(n,i)$ is the set of the sessions on link $(n,i)$. $t_{slot}$ is the duration of a time slot. $C_{ni}(t)$ denotes the transmission capacity of link $(n,i)$ in time slot t. $\mu_{ni}^{(m)}(t)$ is the number of transmissions of packets of session m on link $(n,i)$ in time slot t.

### 2.2. Virtual Queue at The Transport Layer

The number of packets generated by session $m$ in time slot $t$ is denoted by $A_m(t)$. We assume that $A_m(t)$ is limited by $A_{\max}^{(m)}$, which is the maximum allowable number of packets generated by session $m$ in a time slot. $r_m(t) \in [0, A_m(t)]$ is the number of packets of session $m$ injected into the network layer in time slot $t$. $\eta_m(t) \in [0, A_{\max}^{(m)}]$ is the auxiliary variable which is used to limit the lower bound of $r_m(t)$. The source node of each session maintains a virtual queue. The virtual queue at the source node of session $m$ is denoted by $Y_m$, which is updated as:

$$Y_m(t+1) = \max\{Y_m(t) - r_m(t), 0\} + \eta_m(t) . \tag{7}$$

If $Y_m$ is guaranteed to be stable, $\overline{\eta_m} \le \overline{r_m}$ can also be guaranteed [45]. Therefore, when $Y_m$ is stable, $\overline{\eta_m}$ can be used to limit the lower bound of $\overline{r_m}$.

### 2.3. Data Queue at The Network Layer

The sessions in the wireless multimedia sensor networks have different end-to-end QoS requirements. As traditional Diffserv provides a lack of granularity for QoS guaranteed services and cannot adaptively distribute transmission priorities according to the end-to-end QoS requirements and the real-time end-to-end situation of sessions, this paper utilized a Differentiated Queueing Services (DQS) [18] scheme to maintain data queues at the network layers of nodes.

According to the DQS scheme, each node maintains one data queue, which contains all data packets at the network layer. For a packet transmitted in the wireless sensor network, $D$ denotes the maximum allowable end-to-end delay of the packet. Obviously, the maximum allowable end-to-end delay of a real-time application packet is higher than that of a non-real-time application packet. $d$ denotes the actual delay that the packet experiences on the path. $T$ represents the remaining lifetime of the packet. Thus, the following equality is set up:

$$T = D - d . \tag{8}$$

When the packet arrives at a node, if $T$ of the packet is a positive number, the packet is admitted to the data queue, and its queueing sequence in the queue is decided according to the $T$ of the packet. The smaller the $T$ of the packet is, the closer to the head of the queue the packet is arranged. This implies that packets will receive differentiated services according to their end-to-end delay requirements.

The queue backlog at the network layer of node $n$ in time slot $t$ is denoted by $Q_n(t)$, which is updated as:

$$Q_n(t+1) = \max[Q_n(t) - \sum_{i \in O(n)} \mu_{ni}(t), 0] + \sum_{j \in I(n)} \mu_{jn}(t) + \sum_{m \in M} 1_{\{n = s_m\}} \cdot r_m(t) , \tag{9}$$

where $\mu_{ni}(t)$ is the amount of packets forwarded from node $n$ to node $i$ in time slot $t$. $\mu_{ni}(t)$ is constrained by $\mu_{ni}(t) \in [0, \mu_{\max}]$, where $\mu_{\max}$ denotes the maximum amount of packets that can be transmitted between two nodes in a time slot. $1_{\{n = s_m\}}$ is an indicator function which is denoted as 1 if $n = s_m$ and 0 otherwise. Similar to (6), the following inequality can be derived:

$$0 \le \mu_{ni}(t) \le C_{ni}(t) \cdot t_{slot} . \tag{10}$$

$C_{ni}(t)$ denotes the transmission capacity of the link $(n,i)$ in time slot $t$.

### 2.4. Design of Energy Queue

Each node maintains an energy queue that relates to the residual energy amount. The energy queue size of node $n$ in time slot $t$ is $E_n(t)$, which is updated as:

$$E_n(t+1) = E_n(t) - \sum_{i \in O(n)} e_{ni}^T(t) - e_n^{sense}(t) , \tag{11}$$

where $e_{ni}^T(t)$ represents the energy consumption for data transmission from node $n$ to node $i$ in time slot $t$. Here, the superscript $T$ denotes *Transmission*. $t_{ni}^T(t)$ denotes the duration of the data transmission between node $n$ and node $i$ in time slot $t$. Given that the duration of the data transmission between any two nodes in a time slot cannot be greater than the duration of the time slot, the following inequality can be derived:

$$0 \le t_{ni}^T(t) \le t_{slot} . \tag{12}$$

Under the assumption that the time for scheduling and routing is much shorter than the data transmission duration, we can derive:

$$e_{ni}^T(t) = p_{ni}^T(t) \cdot t_{ni}^T(t) \approx p_{ni}^T(t) \cdot t_{slot} . \tag{13}$$

$e_n^{sense}(t)$ is the energy consumption in control packets sensing/processing by node $n$ in time slot $t$. In this paper, we establish that the packets sensing/processing power of each node is a constant $p^{sense}$. The following equality can be derived:

$$e_n^{sense}(t) = p^{sense} \cdot t_{slot} . \tag{14}$$

The initial energy in $E_n$ is set to be $\theta_n^E$, and $\theta_n^E - E_n(t)$ is the energy consumption of node $n$ in time slot $t$.

## 3. Problem Formation and Dynamic Algorithm

*3.1. Design Throughput Utility Optimization Problem*

The traditional optimization problem for cross-layer control is to maximize the sum utilities, which is designed as:

$$\begin{aligned} \text{maximize} \quad & \sum_{m \in M} U_m(r_m) \\ \text{subject to} \quad & \bar{r} \in \Lambda , \end{aligned} \tag{15}$$

where $\Lambda$ represents the capacity region of the network. $\bar{r} = (\bar{r_m}, m \in M)$ denotes the time average throughput of session $m$. The constraint means that the network stability is guaranteed.

However, (15) seeks to maximize the total utilities and cause extreme utility unfairness. (15) is also not applicable to real-time services. Aiming at this issue, the "pseudo utility" function $U_m(r_m)$ [46] was used instead of the utility function $U_m(r_m)$ in the optimization problem. $U_m(r_m)$ is defined as:

$$U_m(r_m) = \int_{r_L}^{r_m} \frac{1}{U_m(y)} dy, \quad r_L \le r_m \le r_H , \tag{16}$$

where $r_L \ge 0$ denotes the minimum transmission rate required by service $m$. $r_H \le \infty$ is the maximum transmission rate required by service $m$.

Following the design principle of the utility function in [46], the utility function $U_m(r_m)$ for real-time services will be given by a sigmoid function as follows:

$$U_m(r_m) = \begin{cases} 0 & if \ r_m \le B_{\min} \\ \dfrac{1}{1 + e^{-a(r_m - b)}} & if \ B_{\min} \le r_m \le B_{\max} \\ 1 & if \ r_m \ge B_{\max} \end{cases} , \tag{17}$$

where $B_{\min}$ and $B_{\max}$ are the minimum and maximum bandwidth constraints on the sigmoid. $a$ is used to control the slope of the sigmoid. $b$ is set to be $\dfrac{B_{\max} - B_{\min}}{2} + B_{\min}$.

The utility function $U_m(r_m)$ for non-real-time services is designed as:

$$U_m(r_m) = \log(r_m + 1) .\tag{18}$$

Obviously, $U_m(r_m)$ designed in (18) and $U_m(r_m)$ designed in (17) with $r_m \in [r_L, r_H]$ are both positive ascending functions.

The utility-fair optimization problem is defined as:

$$\begin{aligned}\text{maximize} \quad & \sum_{m \in M} U_m(r_m) \\ \text{subject to} \quad & \overline{r} \in \Lambda .\end{aligned}\tag{19}$$

The derivative of $U_m(r_m)$: $U'_m(r_m) = \dfrac{1}{U_m(r_m)}$. As $U_m(r_m)$ is a positive ascending function, we can get $U'_m(r_m) > 0$ and $U''_m(r_m) < 0$. Therefore, the optimization problem (19) is a convex optimization problem.

$\overline{\eta_m}$ can be used to limit the lower bound of $\overline{r_m}$, which is introduced in Section 2.2. The utility-fair optimization problem *P1* in this paper is defined as follows:

$$\begin{aligned}\text{maximize} \quad & 1 / B_F^2 \cdot \sum_{m \in M} U_m(\overline{\eta_m}) \\ \text{subject to} \quad & \overline{r} \in \Lambda, \\ & 0 \leq \overline{r} \leq \lambda, \\ & \overline{\eta} \leq \overline{r}, \\ & (1), (2)\end{aligned}\tag{20}$$

where $B_F$ represents the size of packet buffer in each node of the sensor network.

### 3.2. Design Dynamic Algorithm via Lyapunov Optimization

The Lyapunov optimization technique is applied to solve optimization problem *P1*. $Q_n(\forall n \neq d_*)$, $E_n(\forall n \neq d_*)$, and $Y_m(\forall m \in M)$ are used in the dynamic algorithm. Let $\Theta(t) = [Q(t), E(t), Y(t)]$ be the network state vector in time slot *t*, and the Lyapunov function is defined as:

$$L(\Theta(t)) = \frac{1}{2} \Big[ \sum_{m \in M} (Y_m(t) / B_F)^2 + \sum_{n \neq d_*} q_n \tau_n(t) (Q_n(t) / B_F)^2 + \sum_{n \neq d_*} e_n [(\theta_n^E - E_n(t)) / \theta_n^E]^2 \Big] .\tag{21}$$

Normalization of different types of queues is utilized for a unified dimension in the Lyapunov optimization framework. In addition, in Equation (21), there are three coefficients, $q_n$, $e_n$, and $\tau_n(t)$, that are designed as follows.

In order to relay data to the sink node, the energy of the nodes around the sink node will be easily exhausted. When the nodes around the sink node die out, there is no packet that can be received by the sink node, which means the wireless sensor network is out of action. Therefore, the maximal lifetime of the nodes around the sink node decide the network lifetime. To prolong the network lifetime, nodes around the sink node should reduce transmission power, which may result in the increase of relay hops to the sink node and the end-to-end delay. In this paper, $e_n$, which is the distance coefficient of the energy queue in node *n*, is used as the weight of queue $E_n$ in the Lyapunov function. The design principle of $e_n$ is that the nearer node *n* is to the sink node, the higher the weight of $E_n$ will be. $e_n$ is defined as:

$$e_n = e^{\max[0, R - Dis_n]/R} ,\tag{22}$$

where $Dis_n$ denotes the distance between node $n$ and the sink node, $R$ represents the threshold distance, and $e$ is the Euler number. Obviously, (22) fits the design principle of $e_n$.

In the wireless sensor networks, the nodes far from the sink node should be dedicated to increasing the average transmission rates to reduce the end-to-end delay of the packets buffered in them, with little consideration of energy consumption in data transmission. $q_n$ is used to indicate the weight of queue $Q_n$ of node $n$ in the Lyapunov function. $q_n$ is defined as:

$$q_n = e^{\max[0, Dis_n - R]/(Dis_{max} - R)} ,\tag{23}$$

where $Dis_{max}$ denotes the maximal distance between the sink node and any other node in the network. The design principle of $q_n$ is that, when the distance between node $n$ and the sink node is longer than the threshold distance, then the farther node $n$ is to the sink node, the higher the weight of $Q_n$.

$\tau_n(t)$ is the delay coefficient of $Q_n$ in time slot $t$. The design principle of $\tau_n(t)$ is that, the lower the total remaining lifetime of packets in $Q_n(t)$ is, the higher the scheduling priority of $Q_n$. $\tau_n(t)$ is calculated as:

$$\tau_n(t) = e^{\sum_{k \in K_n} (D_k - T_k(t)) / \sum_{k \in K_n} D_k} \cdot 1/e ,\tag{24}$$

where $K_n$ is the set of data packets in node $n$. $D_k$ is defined as the maximum allowable end-to-end delay of packet $k$. $T_k(t)$ denotes the remaining lifetime of packet $k$ in time slot $t$.

The conditional Lyapunov drift in time slot $t$ is:

$$\Delta(\Theta(t)) = E\{L(\Theta(t+1)) - L(\Theta(t)) \mid \Theta(t)\} .\tag{25}$$

To maximize a lower bound of $1/B_F^2 \cdot \sum_{m \in M} U_m(\overline{\eta_m})$, the drift-plus-penalty function can be defined as:

$$\Delta_V(\Theta(t)) = \Delta\Theta(t) - V \cdot E\{1/B_F^2 \cdot \sum_{m \in M} U_m(\eta_m(t)) \mid \Theta(t)\} ,\tag{26}$$

where $V$ is the weight of the utility defined by the user. The following inequality can be derived:

$$\begin{aligned}
E\{\Delta_V \Theta(t)\} \le{}& B' - (1/B_F^2) \cdot \sum_{m \in M} [V \cdot U_m(\eta_m(t)) - Y_m(t) \cdot \eta_m(t)] \\
&- (1/B_F^2) \cdot \sum_{m \in M} r_m(t) \cdot [Y_m(t) - q_n \tau_n(t) \cdot Q_n(t) \cdot 1_{\{n = s_m\}}] \\
&- (1/B_F^2) \cdot \sum_{n \ne d_*} \sum_{i \in O(n)} \mu_{ni}(t) [q_n \tau_n(t) \cdot Q_n(t) - q_i \tau_i(t) \cdot Q_i(t)] \\
&- \sum_{n \ne d_*} (e_n/(\theta_n^E)^2) \cdot (E_n(t) - \theta_n^E) \cdot (\sum_{i \in O(n)} p_{ni}^T(t) \cdot t_{slot} + e^{sense}) .
\end{aligned}\tag{27}$$

In (27), $B''$ is a constant which can be evaluated as:

$$B' = \frac{1}{2}(e_{max}/(\theta_n^E)^2) \cdot (|N| \cdot p_{n,max}^T \cdot t_{slot} + e^{sense})^2 + 2(1/B_F^2) \cdot (A_{max}^{(m)})^2 + \frac{1}{2}(q_{max}\tau_{max}/B_F^2) \cdot (2\mu_{max} + A_{max}^{(m)})^2 \tag{28}$$

where $|N|$ is the number of nodes in the network. $e_{max}$, $\tau_{max}$, and $q_{max}$ are the maximal values of $e_n$, $\tau_n(t)$, and $q_n$. According to (22), (23), and (24), the values of $e_{max}$, $\tau_{max}$ and $q_{max}$ are $e$, $1$, and $e$.

The main design principle of the algorithm CLADS (cross-layer algorithm for different services) is to minimize the right-hand side of (27). The algorithm includes four components:

- **Source rate control:**

For each session $m \in M$ at source node $s_m$, the admitted rate $r_m(t)$ is chosen to solve:

$$\text{maximize} \quad r_m(t) \cdot [Y_m(t) - q_n \tau_n(t) \cdot Q_n(t) \cdot 1_{\{n=s_m\}}]$$

$$\text{subject to} \quad 0 \le r_m(t) \le A_m(t).$$

(29)

Problem (29) is a linear optimization problem, and $r_m(t)$ can be calculated as:

$$r_m(t) = \begin{cases} A_m(t) & \text{if } Y_m(t) > q_n \tau_n(t) \cdot Q_n(t) \cdot 1_{\{n=s_m\}} \\ 0 & \text{if } Y_m(t) \le q_n \tau_n(t) \cdot Q_n(t) \cdot 1_{\{n=s_m\}} \end{cases}.$$

(30)

- **Virtual input rate control:**

For each session $m \in M$ at source node $s_m$, the virtual input rate $\eta_m(t)$ is chosen to solve:

$$\text{maximize} \quad V \cdot U_m(\eta_m(t)) - Y_m(t) \cdot \eta_m(t)$$

$$\text{subject to} \quad 0 \le \eta_m(t) \le A_{\max}^{(m)}.$$

(31)

Since $U_m(.)$ is differentiable according to the definition in (16), considering the constraint of (31), $\eta_m(t)$ can be calculated as:

$$\eta_m(t) = \max[\min[U_m^{-1}(V / Y_m(t)), A_{\max}^{(m)}], 0].$$

(32)

When session *m* is a non-real-time service, the utility function $U_m(t)$ is defined as (18), and the following equality can be derived:

$$U_m^{-1}(V / Y_m(t)) = e^{V/Y_m(t)} - 1.$$

(33)

The utility function of a real-time service is defined as (17), and the following can be derived:

$$U_m^{-1}(V / Y_m(t)) = \begin{cases} b - 1 / a \cdot \log(Y_m(t) / V - 1) & \text{if } Y_m(t) > V \\ B_{\max} & \text{if } Y_m(t) \le V \end{cases}.$$

(34)

- **Joint routing scheduling and power control:**

At each node $n \ne d_*$, the scheme is implemented to route the packets, allocate transmission opportunities to links, and decide the transmission powers. The joint optimization problem of routing, scheduling, and power control is as follows:

$$\text{maximize} \quad \Psi$$

$$\text{subject to} \quad (1), (2), (10).$$

(35)

where the expression of $\Psi$ is as:

$$\Psi = (1 / B_F^2) \cdot \sum_{n \ne d_*} \sum_{i \in O(n)} \mu_{ni}(t)[q_n \tau_n(t) \cdot Q_n(t) - q_i \tau_i(t) \cdot Q_i(t)] + \sum_{n \ne d_*} (e_n / (\theta_n^E)^2) \cdot (E_n(t) - \theta_n^E) \cdot (\sum_{i \in O(i)}$$

(36)

We define $w_{ni}$ as the weight value of link $(n,i)$, that is calculated as:

$$w_{ni} = \max[q_n \tau_n(t) \cdot Q_n(t) - q_i \tau_i(t) \cdot Q_i(t), 0].$$

(37)

It is assumed that the whole transmission capacity of any link will be devoted to data transmission on the link. It is also assumed that in any time slot, the negotiation time is much shorter than the data transmission time. Based on these two assumptions, the following can be derived:

$$\mu_{ni}(t) = C_{ni}(t) \cdot t_{ni}^T(t) \approx C_{ni}(t) \cdot t_{slot}$$

(38)

We define $\delta_t = (e_n / (\theta_n^E)^2) \cdot (E_n(t) - \theta_n^E)$. According to (3) and (5), (35) can be transformed into:

$$\text{maximize} \quad \sum_{n \neq d_*} \sum_{i \in O(n)} \Psi_1 \tag{39}$$

$$\text{subject to} \quad (1), (2),$$

where

$$\Psi_1 = [(1 / B_F^2) \cdot B \cdot \log \left( \frac{G_{ni} p_{ni}^T(t) K_{ni}}{\sum\limits_{k \neq i, k \neq n} G_{ki} p_{kk'}^T(t) + n_i} \right) \cdot w_{ni} + \delta_t \cdot p_{ni}^T(t) . \tag{40}$$

$I_c \in I$ is defined as a link set for simultaneous data transmission. $I$ denotes the set of $I_c$. (39) can be transformed as follows.

$$\text{maximize} \quad \Psi_2(p^T(t)) \tag{41}$$

$$\text{subject to} \quad 0 \leq p_{ni}^T(t) \leq p_{n,\max}^T, n \neq d_*, i \in O(n), (n,i) \in I_c,$$

where

$$\Psi_2(p^T(t)) = \sum_{n \neq d_*, i \in O(n), (n,i) \in I_c, I_c \in I} [(w_{ni} / B_F^2) \cdot B \cdot \log \left( \frac{G_{ni} p_{ni}^T(t) K_{ni}}{\sum\limits_{k \neq i, k \neq n} G_{ki} p_{kk'}^T(t) + n_i} \right) + \delta_t \cdot p_{ni}^T(t)] . \tag{42}$$

**Theorem 1.** *Optimization problem (41) is a log-convex optimization problem. It implies that if we set* $P^T(t) = \log p^T(t)$, *the new optimization problem (43) is derived:*

$$\text{maximize} \quad \Psi_3(P^T(t)) \tag{43}$$

$$\text{subject to} \quad 0 \leq P_{ni}^T(t) \leq \log p_{n,\max}^T, n \neq d_*, i \in O(n), (n,i) \in I_c.,$$

*where*

$$\Psi_3(P^T(t)) = \sum_{n \neq d_*, i \in O(n), (n,i) \in I_c, I_c \in I} [(w_{ni} \cdot B / B_F^2) \cdot \log(G_{ni} K_{ni} e^{P_{ni}^T(t)}) - (w_{ni} \cdot B / B_F^2) \cdot \log(\sum_{k \neq i, k \neq n} G_{ki} e^{P_{kk'}^T(t)} + n_i) + \delta_t \cdot e^{P_{ni}^T(t)}] \tag{44}$$

*The optimization problem (43) is a convex optimization problem.*

**Proof.** For problem (43), if the optimization objective function $\Psi_3(P^T(t))$ is a convex or concave function in regard to $P^T(t)$, and the constraint is a convex set of $P^T(t)$, then (43) is a convex optimization problem.

According to the principle that the logarithm of the sum of $e$ exponent functions is a convex function, then [47] and the conclusion from [48], $(w_{ni} \cdot B / B_F^2) \cdot [\log(G_{ni} K_{ni}) + P_{ni}^T(t) - \log(\sum_{k \neq i, k \neq j} G_{ki} e^{P_{kk'}^T(t)} + n_i)]$, are concave functions. In each time slot, for $E_n(t) \leq \theta_n^E = E_n(0)$, we can get $\delta_t \leq 0$. If the exponent function is convex [48], $\delta_t \cdot e^{P_{ni}^T(t)}$ is a concave function if $\delta_t < 0$. When $\delta_t = 0$, the value of $\delta_t \cdot e^{P_{ni}^T(t)}$ is zero. According to the principle that the sum of concave functions is a concave function, the optimization objective function of (43) is a concave function.

As the constraint of (43) is linear, it is a convex set of $P^T(t)$. Therefore, problem (43) is a convex optimization problem and problem (41) is a log-convex optimization problem. □

In the case of $I_c \in I$ being known, problem (43) can be solved using the interior point method [48]. The optimal transmission powers can then be gained through equality $p^T(t) = e^{P^T(t)}$. However,

the interior point method is not suitable for wireless sensor networks, as this method may result in heavy control message overhead.

　　　Two distributed optimization algorithms (Algorithm 1 and Algorithm 2) can be used to solve problem (41).

---

**Algorithm 1 (Distributed gradient projection power control algorithm)** [49]:

There are $s$ links in link set $I_c$, which are $(n_1, n_1'), (n_2, n_2'), \ldots, (n_s, n_s')$. In each time slot, the optimal transmission power of node $n$ is gained through multiple iterations of $p_{nn'}^T(t)$. $p_{nn'}^T(tt)$ denotes the $tt^{th}$ iteration of $p_{nn'}^T(t)$. $(j, j')$ represents other links, except for $(n, n')$, in the network. $p_{nn'}^T(t)$ is updated as follows:

$$p_{nn'}^T(tt+1) = p_{nn'}^T(tt) + \kappa \cdot \nabla_{(n,n')} \Psi(p^T(tt)),$$
(45)

where $\kappa$ is the step size. The gradient of $\Psi(p^T(tt))$ is calculated as:

$$\nabla_{(n,n')} \Psi(p^T(tt)) = \frac{B \cdot w_{nn'}}{B_F^2 \cdot p_{nn'}^T(tt)} + \delta_t - \sum_{j \neq n} \frac{\frac{B}{B_F^2} \cdot w_{jj'} \cdot G_{nj'}}{\sum_{k \neq j, k \neq j'} (G_{kj'} \cdot p_{kj'}^T(tt) + n_{j'})}.$$
(46)

If we set

$$m_j(tt) = \frac{\frac{B}{B_F^2} \cdot w_{jj'} \cdot SINR_j(tt)}{p_{jj'}^T(tt) \cdot G_{jj'}},$$
(47)

then (45) can be transformed into:

$$p_{nn'}^T(tt+1) = p_{nn'}^T(tt) + \kappa \cdot \left[ \frac{B \cdot w_{nn'}}{B_F^2 \cdot p_{nn'}^T(tt)} + \delta_t - \sum_{j \neq n} G_{nj'} \cdot m_j(tt) \right].$$
(48)

$H$ denotes the Hessian matrix of $\Psi(p^T(tt))$. $l$ is the row-coordinate of $H$, and $c$ is the column-coordinate of $H$. The following equality can be derived:

$$H_{l,c} = \begin{cases} -\dfrac{B \cdot w_{ll'}}{B_F^2 \cdot (p_{ll'}^T(tt))^2} + \displaystyle\sum_{j \neq l} \dfrac{B \cdot w_{jj'} \cdot G_{lj'}^2 / B_F^2}{(\sum_{k \neq j} G_{kj'} \cdot p_{kj'}^T(tt) + n_{j'})^2} & if\ l = c \\[6mm] \displaystyle\sum_{j \neq l, j \neq c} \dfrac{B \cdot w_{jj'} \cdot G_{lj'} \cdot G_{cj'} / B_F^2}{(\sum_{k \neq j} G_{kj'} \cdot p_{kj'}^T(tt) + n_{j'})^2} & if\ l \neq c \end{cases}.$$
(49)

$L$ and $C$ are defined as:

$$L = \max_l \left[ \sum_{c, c \neq l} \sum_{j \neq l, j \neq c} \frac{B \cdot w_{jj'} \cdot G_{lj'} \cdot G_{cj'} / B_F^2}{(\sum_{k \neq j} G_{kj'} \cdot p_{kj'}^T(tt) + n_{j'})^2} + \left| -\frac{B \cdot w_{ll'}}{B_F^2 \cdot (p_{ll'}^T(tt))^2} + \sum_{j \neq l} \frac{B \cdot w_{jj'} \cdot G_{lj'}^2 / B_F^2}{(\sum_{k \neq j} G_{kj'} \cdot p_{kj'}^T(tt) + n_{j'})^2} \right| \right]$$
(50)

$$C = \max_c \left[ \sum_{l, l \neq c} \sum_{j \neq l, j \neq c} \frac{B \cdot w_{jj'} \cdot G_{lj'} \cdot G_{cj'} / B_F^2}{(\sum_{k \neq j} G_{kj'} \cdot p_{kj'}^T(tt) + n_{j'})^2} + \left| -\frac{B \cdot w_{cc'}}{B_F^2 \cdot (p_{cc'}^T(tt))^2} + \sum_{j \neq c} \frac{B \cdot w_{jj'} \cdot G_{cj'}^2 / B_F^2}{(\sum_{k \neq j} G_{kj'} \cdot p_{kj'}^T(tt) + n_{j'})^2} \right| \right]$$
(51)

---

　　　We set $K = \sqrt{L \cdot C}$, $\varepsilon \leq \dfrac{2}{1+K}$. The step size $\kappa$ is set as: $\varepsilon \leq \kappa \leq \dfrac{2-\varepsilon}{K}$. If step size is small enough, the algorithm can achieve global convergence.

The details of the algorithm implemented in each iteration $tt$ are as follows: (i) The receiving node of each link $(j, j')$ in $I_c$ measure and gain $SINR_j(tt)$ and $G_{jj'}$. Node $j$ also calculates $w_{jj'}$ and $m_j(tt)$ using (37) and (47), respectively. (ii) Each receiving node $j$ broadcasts $m_j(tt)$ in the network. (iii) After gathering $m_j(tt)$ from all receiving nodes, each send node updates its transmission power according to (48). (iv) $tt = tt + 1$, and jump to (i) again. (v) The iterations are stopped until convergence is achieved.

---

**Algorithm 2 (Block coordinate descent power control algorithm) [41]:**

There are $s$ links in link set $I_c$, which are $(n_1, n_1'), (n_2, n_2'), ..., (n_s, n_s')$. There are multiple iterations in a time slot. In each iteration, the optimal transmission powers of links in $I_c$ are calculated in turn. The optimization process in each iteration is as follows:

$$P_{n_1 n_1'}^T(tt+1) = \underset{P_{n_1 n_1'}^T}{\arg\max}\ \Psi(P_{n_1 n_1'}^T, P_{n_2 n_2'}^T(tt), P_{n_3 n_3'}^T(tt), ..., P_{n_s n_s'}^T(tt))$$

$$P_{n_2 n_2'}^T(tt+1) = \underset{P_{n_2 n_2'}^T}{\arg\max}\ \Psi(P_{n_1 n_1'}^T(tt+1), P_{n_2 n_2'}^T, P_{n_3 n_3'}^T(tt), ..., P_{n_s n_s'}^T(tt))$$

$$......$$

$$P_{n_s n_s'}^T(tt+1) = \underset{P_{n_s n_s'}^T}{\arg\max}\ \Psi(P_{n_1 n_1'}^T(tt+1), P_{n_2 n_2'}^T(tt+1), P_{n_3 n_3'}^T(tt+1), ..., P_{n_s n_s'}^T).$$

---

Iterations are stopped until convergence is achieved. As (43) is a convex optimization problem, the BCD algorithm can achieve global convergence. The optimal solutions of (41) can be obtained through equality $p^T(t) = e^{P^T(t)}$.

In essence, the above central joint routing, scheduling, and power control algorithm for (35) is an MWM (maximal weighted matching) algorithm [50]. The computational complexity to obtain link set $I_c \in I$ is $O(|N|^3)$. $|N|$ is the number of nodes in the network. The increase of $|N|$ will lead to a sharp rise in the computational complexity. In order to reduce the computational complexity, the distributed media access control scheme proposed in section V of [51] can be used instead of the optimal central algorithm. The computational complexity of the distributed scheme at each node is lower than $O(|N||L|^2)$, where $|L|$ is the number of existing links in the network [51]. Under the distributed scheme, the transmission power of node $n$ can be set to $p_{n,\max}^T$. Nodes with higher weight values can obtain transmission opportunities with higher probabilities. The distributed scheme is essentially a GMS (greedy maximal scheduling) algorithm. As the capacity region of GMS can reach half the capacity region of MWM, the capacity region of the distributed media access control scheme can also reach half the optimal capacity region. Here, the capacity region of a policy is defined as the collections of all traffic load matrices which are sustainable by the specific policy [44].

- **Update of queues:**

$Y(t)$, $Q(t)$ and $E(t)$ are updated using (7), (9), and (11) in each time slot.

## 4. Simulation

### 4.1. Simulation Setup

The topology was a multi-hop wireless sensor network with several nodes randomly distributed in a square of 500 m × 500 m. The sink node was in the center of the square. In the simulation, nodes did not move. Each node was aware of the location of other nodes in the network. The messages broadcasted on the common control channel by any node were received by all other nodes. The data transmission scope of a node was 125m. The threshold distance $R$, which was introduced in Section

3.2, was set to be 150m. Data was transmitted in packets with a length of 4000 bits. There was a total of $N$ nodes in the network, including one sink node, $0.8 \cdot N$ non-real-time nodes, and $0.2 \cdot N - 1$ multimedia nodes. The data buffer sizes of the sink node, non-real-time nodes, and multimedia nodes were $2 \times 10^8$ bits, $2 \times 10^7$ bits and $2 \times 10^7$ bits, respectively, which could also be recorded as 50,000 packets, 5000 packets, and 5000 packets. We assumed that the energy of nodes could only be consumed in data transmission. The available transmission power levels included 2 mW, 5 mW, 8 mW, and 10 mW. The channel bandwidth was 1 MHz. The transmit loss from node $j$ to $i$ was $G_{ji} = \dfrac{1}{(d_{ji})^4}$, where $d_{ji}$ is the distance between node $j$ and node $i$. $K_{ji} = 5$, $n_i = 10^{-13}$ W, $e^{sense} = 0$, $V = 50$.

In the network, the sink node was the destination node of all sessions. There were $0.2 \cdot N - 1$ real-time sessions generated at multimedia nodes. $A_m(t)$ of real-time sessions was set to be 30packets/s. This implies that the data arrival rate of any real-time service in each multimedia node was 120Kbits/s. The end-to-end delay deadline of any real-time session was 1 s. There were $0.8 \cdot N$ non-real-time sessions generated at non-real-time nodes. $A_m(t)$ of non-real-time sessions was 1 packets/s. This implies that the data arrival rate of any non-real-time service in each non-real-time node was 4Kbits/s. The end-to-end delay deadline of any non-real-time session was 4 s. In this simulation, $A_{\max}^{(m)} = A_m(t) + 0.1$.

There were two CLADS-based schemes: CLADS-BCD and CLADS-MAC. In CLADS-BCD scheme, $I_c \in I$ were obtained through enumeration on all link sets, and the BCD algorithm was adopted in power control. In the CLADS-MAC scheme, the distributed media access control scheme proposed in [51] was utilized in scheduling and the transmission power was set to be $p_{n,\max}^T$. In the distributed media access control scheme, $\varphi = 10$, $\gamma = 10$, DIFS = 50 $\mu s$, SIFS = 20 $\mu s$, Dtime = 50 $\mu s$, and minislot = 20 $\mu s$. In one time slot, the maximum allowable number of times for a sending node to send RTS packets to establish links with the receiving node was 3. In both CLADS-BCD and CLADS-MAC, the utility function for real-time services was (17). The parameters of (17) were set as $a = 1$, $B_{\min} = 0$, and $B_{\max} = 30.1$. The utility function for non-real-time services was (18). The performances of the CLADS-BCD scheme and the CLADS-MAC scheme were compared with that of EASYO [41]—a backpressure-style scheme—and ACH [24]—a cluster-based scheme. In EASYO, the utility function for both real-time and non-real-time services was (18).

The time slot duration was 40 ms.

*4.2. Comparison of Performance*

In the simulation of Figures 1–8, performances of CLADS-BCD, CLADS-MAC, EASYO, and ACH are compared. The initial energy in the sink node, non-real-time nodes, and multimedia nodes were 5 J, 0.5 J, and 2 J, respectively. This simulation lasted for 100,000 time slots.

The stability period is defined as the time duration from the establishment of the network to the death of the first node [24]. The stability period is crucial for the applications which require the feedback from the sensor network to be reliable [22]. In Figure 1, the stability periods of different schemes are compared. It can be seen that the stability period of ACH was the longest. The stability period of CLADS-BCD was longer than that of CLADS-MAC and EASYO. Therefore, in backpressure-style schemes, CLADS-BCD makes the network more stable than CLADS-MAC and EASYO. In wireless sensor networks, the energy of the nodes around the sink node can easily be exhausted, which limits the stability periods of WSNs. Under ACH, in each cluster-setup round, the nodes with more residual energy can be chosen as the cluster heads with higher probabilities, and this can balance the energy consumption of nodes around the sink node. Therefore, networks under ACH had the longest stability periods. CLADS-BCD, CLADS-MAC, and EASYO all consider residual energy states of nodes in scheduling to balance energy consumption. In the CLADS schemes, distance coefficients for energy queues are applied to reduce the energy consumption of nodes around the sink node, and this leads to longer stability periods under CLADS-BCD than under EASYO. The reason why the stability periods under CLADS-MAC were the shortest is that CLADS-MAC adopts

a distributed media access control scheme in scheduling, obtaining sub-optimal transmission link sets.

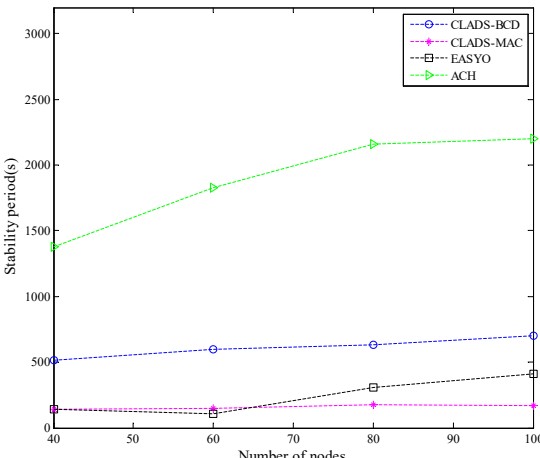

**Figure 1.** Stability period with varying number of nodes.

The consumed powers of the network under different schemes are compared in Figure 2. Figure 2 shows that the consumed power of the network under cluster-based ACH was the lowest. Figure 2 also shows that the consumed power of the network under CLADS-BCD was lower than those under EASYO and CLADS-MAC. The reason is that TDMA scheduling was adopted in each cluster, which effectively reduces consumed power. Unlike EASYO, in CLADS-BCD, queues were normalized in a Lyapunov function to unify the dimension of the queues, which led to more valid weights of the energy queues in the framework, and a reduction in consumed power. The transmission link sets obtained by the distributed media access control scheme in the scheduling of CLADS-MAC were sub-optimal, and this increased the consumed power of the network.

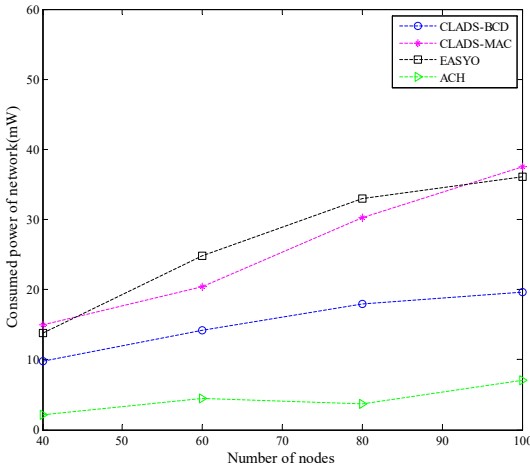

**Figure 2.** Consumed power of the network with varying number of nodes.

In Figure 3 and Figure 4, the ratios of survival nodes in and out of the region of radius $R$ are compared, respectively. From Figure 3 and Figure 4, we can see that ACH performed best most of the time. The reason is that under ACH, energy consumption is well balanced, due to the use of clusters. TDMA scheduling is also utilized in clusters to further reduce energy consumption. The ratios of survival nodes both in and out of the region of radius $R$ under CLADS-BCD were much higher than those under CLADS-MAC and EASYO. This implies that, for backpressure-style schemes,

the network lifetime under CLADS-BCD is longer than that under EASYO and CLADS-MAC. The reason is as follows. Transmission powers of the nodes around the sink node are reduced through the utilization of distance coefficients for energy queues and distance coefficients for data queues in the CLADS schemes. Therefore, the average lifetime of nodes in the region of radius *R* under CLADS-BCD is greatly increased. CLADS also performs better than EASYO in reducing the energy consumption of nodes out of the region of radius *R*, through applying the normalization of queues in optimization framework. In addition, since CLADS-BCD is optimal and CLADS-MAC is sub-optimal in scheduling, CLADS-BCD performs better than CLADS-MAC.

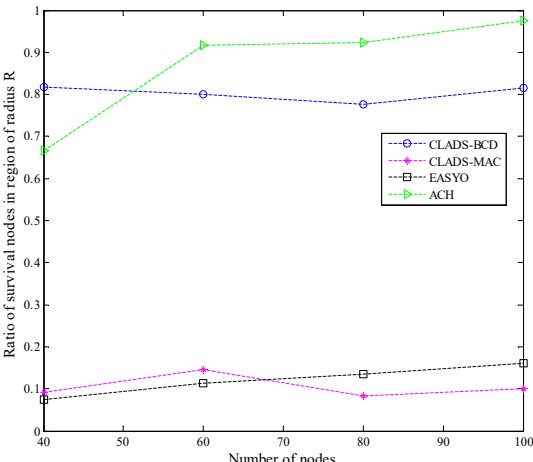

**Figure 3.** Comparison of the ratio of survival nodes in the region of radius R.

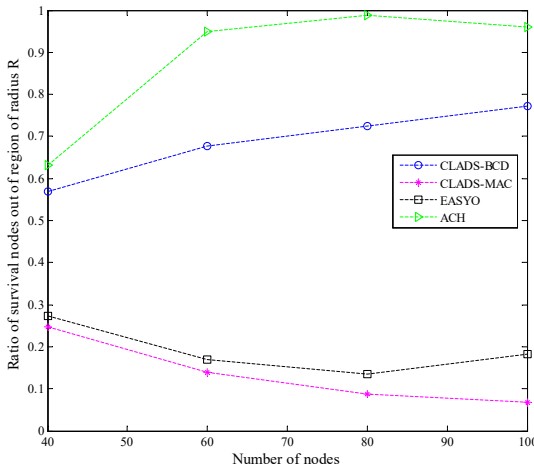

**Figure 4.** Comparison of the ratio of survival nodes out of region of radius R.

The average throughput of real-time sessions of different schemes is compared in Figure 5. The average throughput of real-time sessions under CLADS-BCD, CLADS-MAC, and EASYO all increased with the increase in the number of nodes. In Figure 5, we can see that average throughput of CLADS-BCD was lower than that of EASYO. ACH performed much worse than backpressure-style schemes. The average throughput of non-real-time sessions of different schemes is compared in Figure 6. In Figure 6 we can see that the average throughput of non-real-time sessions under CLADS-BCD, CLADS-MAC, and EASYO all increased with the increase in the number of nodes. Figure 6 also shows that the average throughput of CLADS-BCD was lower than that of EASYO, but higher than that of CLADS-MAC and ACH. The reason is as follows. Both the data transmission time and throughput under ACH were reduced, since TDMA scheduling was utilized in clusters. Since the

weights of data queues in the optimization framework in CLADS-BCD were lower than those in the EASYO scheme, CLADS-BCD and CLADS-MAC performed worse than EASYO in average throughput. It should be noted that the average throughput of real-time sessions under CLADS-MAC was the highest. The reason is that in CLADS-MAC, using a distributed scheduling scheme, the nodes with more buffering packets could obtain more transmission opportunities. This means that CLADS-MAC is suitable for wireless multimedia ad hoc networks in which nodes are initialed with adequate energy.

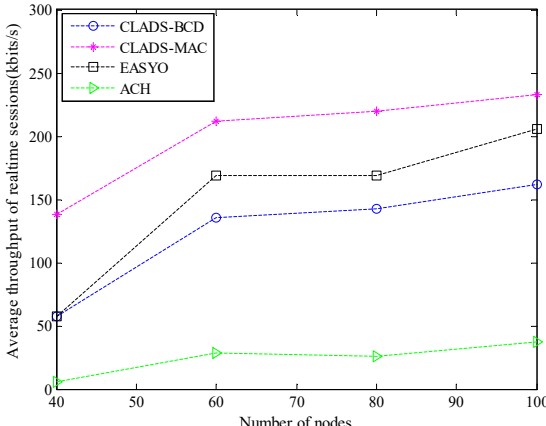

**Figure 5.** Average throughput of real-time sessions under different number of nodes.

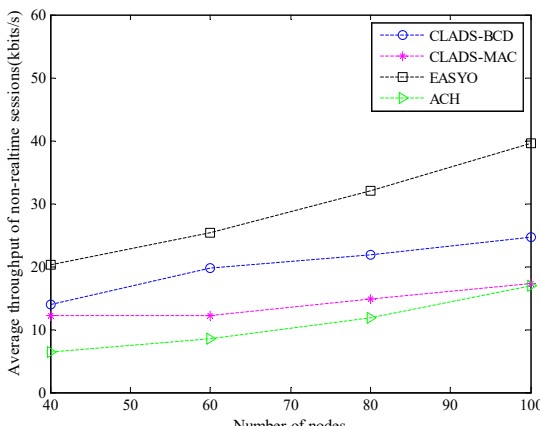

**Figure 6.** Average throughput of non-real-time sessions under different number of nodes.

The average end-to-end delay of real-time and non-real-time sessions of different schemes is compared in Figure 7 and Figure 8, respectively. As Figure 7 shows, CLADS-MAC performed best in the average end-to-end delay of real-time sessions. The reason is that in the scheduling of CLADS-MAC, the nodes with more buffering packets may obtain transmission opportunities with higher probability, and this can reduce average end-to-end delay. According to the results in Figure 5 and Figure 7, CLADS-MAC is the most effective scheme for real-time services in wireless ad hoc networks in which nodes have enough energy. Though the average end-to-end delay of real-time sessions under CLADS-BCD was the highest, it was lower than the end-to-end delay deadlines of real-time sessions, which was 1 s. In Figure 8, the average end-to-end delay of non-real-time sessions under CLADS-BCD was still the highest, and was lower than the end-to-end delay deadline of non-real-time sessions, which was set to be 4 s. This implies that the differences between the average end-to-end delays of various schemes will not lead to differences in the service qualities of applications. The reason is as follows. The weights of the data queues in the Lyapunov optimization framework, as well as the data transmission priorities under CLADS-BCD, were lower than those under EASYO.

Therefore, the average end-to-end delay under CLADS-BCD is higher. However, a DQS scheme is utilized to maintain data queues in CLADS-BCD. In each data queue, a transmission opportunity is distributed to the packet with the lowest survival time. Therefore, CLADS-BCD can provide QoS guarantees on the average end-to-end delay.

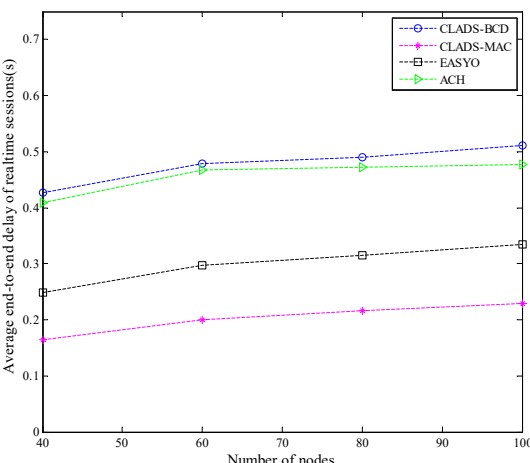

**Figure 7.** Average end-to-end delay of real-time sessions under different numbers of nodes.

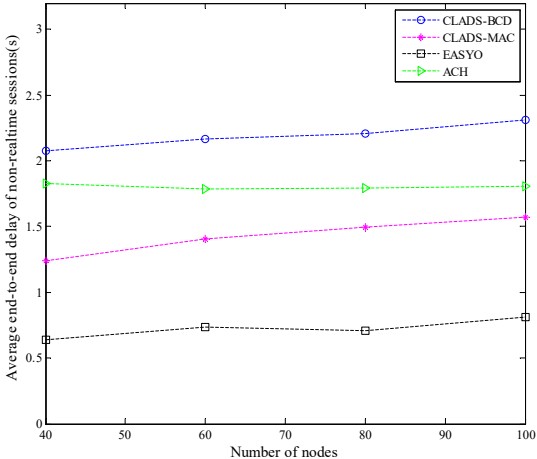

**Figure 8.** Average end-to-end delay of non-real-time sessions under different numbers of nodes.

CLADS and EASYO are both backpressure-based cross-layer algorithms. In Figure 9, Figure 10, and Figure 11, the performances of CLADS-BCD and EASYO are compared. To limit the duration of this simulation, the initial energy of non-real-time nodes and multimedia nodes was set to be 0.01 J and 0.04 J, respectively.

In Figure 9, network lifetime is compared. As shown in the figure, network lifetime under CLADS-BCD was much higher than that under EASYO. The reason is as follows. According to the characteristics of wireless sensor networks, CLADS-BCD adopts distance coefficients of energy queues in optimization to reduce the energy consumption of nodes in the area surrounding the sink node. This can prolong the lifetime of the nodes near the sink node, as well as increase the network lifetime.

The average number of packets of non-real-time and real-time services arriving at the sink node are compared in Figure 10 and Figure 11, respectively. From these two figures we can see that, for both real-time service and non-real-time service, the number of packets arriving at the sink node under CLADS-BCD was higher than the number under EASYO. This implies that wireless sensor

networks using CLADS-BCD can collect more information than networks using EAYSO. The reason is that the network lifetime of CLADS-BCD was much higher than that of EASYO. Thus, the sink node can collect data packets for a much longer time under CLADS-BCD than under EASYO.

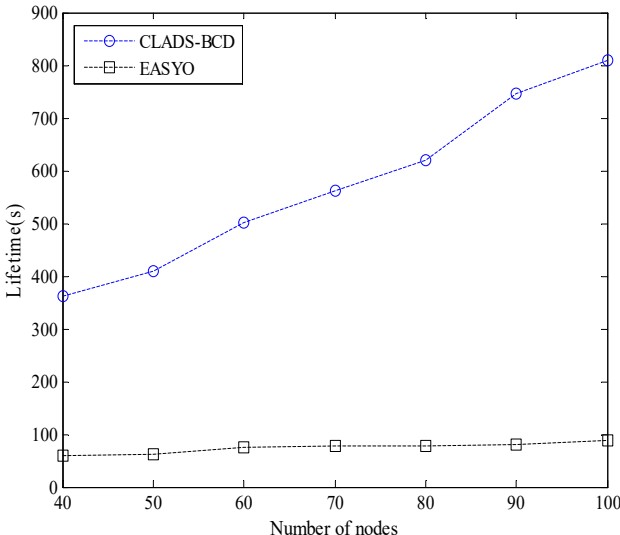

**Figure 9.** Network lifetime.

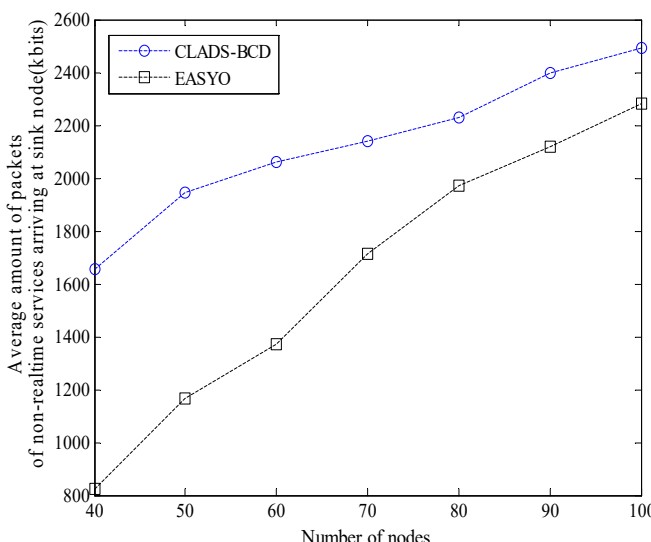

**Figure 10.** Average number of packets of non-real-time services arriving at the sink node.

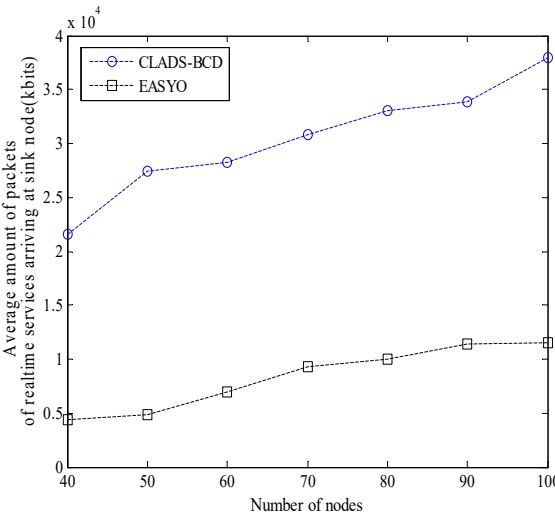

**Figure 11.** Average number of packets of real-time services arriving at the sink node.

The simulations results indicate the conclusions as follows. (i) The network lifetime under ACH is the longest. However, the average throughput under ACH is the lowest. Therefore, ACH is suitable for low-rate wireless sensor networks. (ii) CLADS-MAC is suitable for real-time applications in wireless networks with nodes that have adequate energy. (iii) Though CLADS-BCD performs worse than EASYO in QoS performances, it can obtain a longer network lifetime than EASYO. Thus, networks under CLADS-BCD can collect more data through more working time than networks under EASYO. This indicates that CLADS-BCD achieves the best trade-off between network lifetime and QoS performances in these schemes.

## 5. Conclusions

This paper proposed a cross-layer QoS scheme, which can achieve good a trade-off between QoS performances and network lifetime in wireless multimedia sensor networks. Transmission opportunities are adaptively distributed to different types of applications based on a DQS scheme. By designing a Lyapunov function according to the characteristics of wireless sensor networks, network lifetime is prolonged, with an acceptable reduction in QoS performances. The proposed cross-layer scheme, combined with a low computational complexity distributed media access control scheme, is suitable for real-time services in wireless ad hoc networks. For future studies, we plan to combine this scheme with video transmission in wireless multimedia sensor networks.

**Funding:** This work was supported by the National Natural Science Foundation of China under Grant no. 61603415 and Fundamental Research Funds for the Central Universities of Criminal Investigation Police University of China under Grant no. D2018012.

**Conflicts of Interest:** The authors declare no conflict of interest.

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
