# Peer review of "A Cross-Layer Optimization QoS Scheme in Wireless Multimedia Sensor Networks"

_algorithms, doi:10.3390/a12040068_

Reviewer 1 Report

The reviewer considers that the following issues should be addressed in the work:

1) A more recent bibliographical review that highlights the novelty of the proposal.

2) An experimental analysis of the features of the proposal, since the results shown are only based on simulation.

Author Response

Dear Reviewer:

Thank you very much for giving me an opportunity to revise my manuscript, I appreciate you very much for your positive and constructive comments and suggestions on the manuscript entitled “A Cross-Layer Optimization QoS Scheme in Wireless Multimedia Sensor Networks”. (Manuscript ID: algorithms-440127). Those comments are all valuable and very helpful for revising and improving our paper, as well as the important guiding significance to my researches.

I have studied your comments carefully and have made revisions. I have tried our best to revise the manuscript according to the comments. I sincerely hope this manuscript will be finally acceptable to be published on journal Algorithms.

I would like to express my great appreciation to you for comments on the paper. Looking forward to hearing from you.

The responses to the reviewers’ comments are as following. The order of the responses follows the order of comments in the decision letters.

Point 1:A more recent bibliographical review that highlights the novelty of the proposal.

Response 1: Thanks for the reviewer’s kind advice.

To highlight the novelty of the proposal, seven recent references were added. The new references are marked in yellow in the paper. The summaries of these references are as follows. [2] introduces recent works to develop multimedia applications supported by information-centric networking. [27] and [28] are recent cluster-based energy-efficient schemes in wireless sensor networks. [36] and [37] are recent optimization-based schemes to prolong network lifetime in wireless sensor networks. [42] and [43] are recent backpressure-based cross-layer algorithms that can reduce energy consumption in wireless networks. The list of the new references is as follows.

[2] Muhammad, F. M.; Syed, H. A.; Siraj, M.; Song, H. B.; Danda, B. R. Multimedia streaming in information-centric networking: A survey and future perspectives. Computer Networks. 2017, 125, 103-121.

[27] Peyman, N.; Saeid, A.; Mahmoud, N.; et al. Hierarchical clustering-task scheduling policy in cluster-based wireless sensor networks. IEEE Transactions on Industrial Informatics. 2018, 14, 1876-1886.

[28] Prince, R.; Pragya, D. Optimized and load balanced clustering for wireless sensor networks to increase the lifetime of WSN using MADM approaches. Wireless Networks. 2018, [Online] Available: https://doi.org/10.1007/s11276-018-1812-2

[36] Fabian, C.; Andre, R.; Marc, S.; Nubia, V. An exact approach to extend network lifetime in a general class of wireless sensor networks. Information Sciences. 2017, [Online] Available: https://doi.org/10.1016/j.ins.2017.12.028

[37] Shalli, R.; Syed, H. A.; Rajneesh, T.; Jyoteesh, M.; Song, H. B. IoMT: A reliable cross layer protocol for Internet of multimedia Things. IEEE Internet of Things Journal. 2017, 4, 832-839.

[42] Miguel, C. F.; Carles, A. H.; Javier, M.; et al. Stochastic routing and scheduling policies for energy harvesting communication networks. IEEE Transactions on Signal Processing. 2018, 66, 3363-3376.

[43] Qiu, T.; Qiao, R. X.; Wu, Q. D. P. EABS: an event-aware backpressure scheduling scheme for emergency Internet of Things. IEEE Transactions on Mobile Computing. 2018, 17, 72-84.

To improve quality and highlight the novelty of this manuscript, there are some modifications in the introduction section. The detailed modifications are as follows.

1.      In Line 27 to 31, to show the research value of realtime applications in wireless sensor networks, “Wireless multimedia sensor networks consist of non-realtime nodes which generate non-realtime data and multimedia nodes which are source nodes of audio and video sessions [1].” was replaced by “Realtime applications including multimedia applications have been developed as important services in various types of networks [1][2]. In wireless multimedia sensor networks, realtime applications and non-realtime coexist.”

2.      In Line 62 to 66, to emphasize the features and advantages of backpressure-based schemes, we added “Different from optimization-based schemes, backpressure optimization problems are established based on Lyapunov optimization framework with time being slotted, and the lower bounds of the time average values of optimization utilities achieved by backpressure-based schemes can be pushed arbitrarily to the optimal values[44].”

3.      In Line 68 to 71, to introduce the main content of the new references about backpressure-based schemes, “[36] combines data queues and energy queues in Lyapunov functions for cross-layer control.” is replaced by “In [41] and [42], both data queues and energy queues are utilized in Lyapunov functions for cross-layer control. In [43], the shortest path is combined with backpressure scheme in the process of next-hop node selecting to prolong network lifetime.”

Point 2:An experimental analysis of the features of the proposal, since the results shown are only based on simulation.

Response 2: Thanks for the reviewer’s kind advice. Experimental analysis of the features of the proposal and discussion on the simulation results should be addressed. Experimental analysis for each figure of simulation results was added, and the details are as follows. “”

1.      In Line 458 to 468, for Fig. 1, we added the experimental analysis as “In wireless sensor networks, energy of nodes around the sink node can easily be exhausted which limits the stability periods of WSN. Under ACH, in each cluster-setup round, the nodes with more residual energy can be chosen as the cluster heads with higher probabilities, and this can balance energy consumption of nodes around the sink node. Therefore, networks under ACH have the longest stability periods. CLADS-BCD, CLADS-MAC and EASYO all consider residual energy states of nodes in scheduling to balance energy consumption. In CLADS schemes, distance coefficients for energy queues are applied to reduce energy consumption of nodes around the sink node, and this leads to longer stability periods under CLADS-BCD than under EASYO. The reason why the stability periods under CLADS-MAC are the shortest is that CLADS-MAC adopts distributed media access control scheme in scheduling with obtaining sub-optimal transmission link sets.”

2.      In Line 474 to 479, for Fig. 2, we added the experimental analysis as “The reason is that, in each cluster, TDMA scheduling is adopted, which effectively reduces consumed power. Different from EASYO, in CLADS-BCD queues are normalized in Lyapunov function to unify the dimension of queues, which leads to more valid weights of energy queues in the framework and reduction of consumed power. The transmission link sets obtained by the distributed media access control scheme in scheduling of CLADS-MAC are sub-optimal, and this increases consumed power of network.”

3.      In Line 483 to 485, for Fig. 3 and Fig. 4, we added the experimental analysis as “The reason is that under ACH energy consumption is well balanced using clusters. TDMA scheduling is also utilized in clusters to further reduce energy consumption.”

In Line 488 to 494, for Fig. 3 and Fig. 4, we added the experimental analysis as “The reason is as follows. Transmission powers of nodes around the sink node are reduced through utilization of distance coefficients for energy queues and distance coefficients for data queues in CLADS schemes. Therefore, the average lifetime of nodes in the region of radius R under CLADS-BCD is increased a lot. CLADS also performs better than EASYO in reducing energy consumption of nodes out of the region of radius R through applying normalization of queues in optimization framework. In addition, since CLADS-BCD is optimal and CLADS-MAC is sub-optimal in scheduling, CLADS-BCD performs better than CLADS-MAC.”

4.      In Line 506 to 514, for Fig. 5 and Fig. 6, we added the experimental analysis as “The reason is as follows. Both the data transmission time and throughput under ACH are reduced since TDMA scheduling is utilized in clusters. Since weights of data queues in optimization framework in CLADS-BCD are lower than those in EASYO scheme, CLADS-BCD and CLADS-MAC performs worse than EASYO in average throughput. It should be noted that, average throughput of realtime sessions under CLADS-MAC is the highest. The reason is that in CLADS-MAC using distributed scheduling scheme, the nodes with more buffering packets can obtain more transmission opportunities. This means that, CLADS-MAC is suitable for wireless multimedia Ad hoc networks in which nodes are initialed with adequate energy.”

5.      For Fig. 7 and Fig. 8, in Line 522 to 524, we added the experimental analysis as “The reason is that in scheduling of CLADS-MAC, the nodes with more buffering packets may obtain transmission opportunities with higher probability, and this can reduce average end-to-end delay.”

In Line 531 to 536, we added the experimental analysis as “The reason is as follows. Weights of data queues in Lyapunov optimization framework as well as data transmission priorities under CLADS-BCD are lower than those under EASYO. Therefore, the average end-to-end delay under CLADS-BCD is higher. However, DQS scheme is utilized to maintain data queues in CLADS-BCD. In each data queue, transmission opportunity is distributed to the packet with the least survival time. Therefore, CLADS-BCD can provides QoS guarantees on average end-to-end delay.”

6.      In Line 545 to 549, for Fig. 9, we added the experimental analysis as “The reason is as follows. According to the characteristics of wireless sensor networks, CLADS-BCD adopts distance coefficients of energy queues in optimization to reduce energy consumption of nodes in the surrounding area of the sink node. This can prolong lifetime of nodes near the sink node, as well as increase network lifetime.”

7.      In Line 554 to 556, for Fig. 10 and Fig. 11, we added the experimental analysis as “The reason is that the network lifetime of CLADS-BCD is much higher than that of EASYO. Then, the sink node can collect data packets for much more time under CLADS-BCD than under EASYO.”

Reviewer 2 Report

This paper targets at two main challenges in wireless multimedia sensors networks.

This paper, proposes a joint flow control, routing, scheduling and power control scheme based on Lyapunov optimization framework for increasing the network lifetime and scheduling fairness.

Overall this paper presents a solid and complete work. However, this paper didn't present the state of the art of relevant topics. Some important references are missing. A more comprehensive literature review of relevant topics is desired.

Some example references are: S. Rani, S. H. Ahmed, R. Talwar, J. Malhotra and H. Song, "IoMT: A Reliable Cross Layer Protocol for Internet of Multimedia Things," in IEEE Internet of Things Journal, vol. 4, no. 3, pp. 832-839, June 2017. doi: 10.1109/JIOT.2017.2671460; Muhammad Faran Majeed, et al, Multimedia streaming in information-centric networking: A survey and future perspectives, Computer Networks, Volume 125, 2017, Pages 103-121, ISSN 1389-1286, https://doi.org/10.1016/j.comnet.2017.05.030.

Author Response

Dear Reviewer:

T Thank you very much for giving me an opportunity to revise my manuscript, I appreciate you very much for your positive and constructive comments and suggestions on the manuscript entitled “A Cross-Layer Optimization QoS Scheme in Wireless Multimedia Sensor Networks”. (Manuscript ID: algorithms-440127). Those comments are all valuable and very helpful for revising and improving our paper, as well as the important guiding significance to my researches.

I have studied your comments carefully and have made revisions. I have tried our best to revise the manuscript according to the comments. I sincerely hope this manuscript will be finally acceptable to be published on journal Algorithms.

I would like to express my great appreciation to you for comments on the paper. Looking forward to hearing from you.

The responses to the reviewers’ comments are as following. The order of the responses follows the order of comments in the decision letters.

Point 1: Overall this paper presents a solid and complete work. However, this paper didn't present the state of the art of relevant topics. Some important references are missing. A more comprehensive literature review of relevant topics is desired.

Some example references are: S. Rani, S. H. Ahmed, R. Talwar, J. Malhotra and H. Song, "IoMT: A Reliable Cross Layer Protocol for Internet of Multimedia Things," in IEEE Internet of Things Journal, vol. 4, no. 3, pp. 832-839, June 2017. doi: 10.1109/JIOT.2017.2671460; Muhammad Faran Majeed, et al, Multimedia streaming in information-centric networking: A survey and future perspectives, Computer Networks, Volume 125, 2017, Pages 103-121, ISSN 1389-1286, https://doi.org/10.1016/j.comnet.2017.05.030.

Response 1: Thanks for the reviewer’s kind advice.

To highlight the novelty of the proposal, seven recent references were added. The new references are marked in yellow in the paper. The summaries of these references are as follows. [2] introduces recent works to develop multimedia applications supported by information-centric networking. [27] and [28] are recent cluster-based energy-efficient schemes in wireless sensor networks. [36] and [37] are recent optimization-based schemes to prolong network lifetime in wireless sensor networks. [42] and [43] are recent backpressure-based cross-layer algorithms that can reduce energy consumption in wireless networks. The list of the new references is as follows.

[2] Muhammad, F. M.; Syed, H. A.; Siraj, M.; Song, H. B.; Danda, B. R. Multimedia streaming in information-centric networking: A survey and future perspectives. Computer Networks. 2017, 125, 103-121.

[27] Peyman, N.; Saeid, A.; Mahmoud, N.; et al. Hierarchical clustering-task scheduling policy in cluster-based wireless sensor networks. IEEE Transactions on Industrial Informatics. 2018, 14, 1876-1886.

[28] Prince, R.; Pragya, D. Optimized and load balanced clustering for wireless sensor networks to increase the lifetime of WSN using MADM approaches. Wireless Networks. 2018, [Online] Available: https://doi.org/10.1007/s11276-018-1812-2

[36] Fabian, C.; Andre, R.; Marc, S.; Nubia, V. An exact approach to extend network lifetime in a general class of wireless sensor networks. Information Sciences. 2017, [Online] Available: https://doi.org/10.1016/j.ins.2017.12.028

[37] Shalli, R.; Syed, H. A.; Rajneesh, T.; Jyoteesh, M.; Song, H. B. IoMT: A reliable cross layer protocol for Internet of multimedia Things. IEEE Internet of Things Journal. 2017, 4, 832-839.

[42] Miguel, C. F.; Carles, A. H.; Javier, M.; et al. Stochastic routing and scheduling policies for energy harvesting communication networks. IEEE Transactions on Signal Processing. 2018, 66, 3363-3376.

[43] Qiu, T.; Qiao, R. X.; Wu, Q. D. P. EABS: an event-aware backpressure scheduling scheme for emergency Internet of Things. IEEE Transactions on Mobile Computing. 2018, 17, 72-84.

To improve quality and highlight the novelty of this manuscript, there are some modifications in the introduction section. The detailed modifications are as follows.

1.      In Line 27 to 31, to show the research value of realtime applications in wireless sensor networks, “Wireless multimedia sensor networks consist of non-realtime nodes which generate non-realtime data and multimedia nodes which are source nodes of audio and video sessions [1].” was replaced by “Realtime applications including multimedia applications have been developed as important services in various types of networks [1][2]. In wireless multimedia sensor networks, realtime applications and non-realtime coexist.”

2.      In Line 62 to 66, to emphasize the features and advantages of backpressure-based schemes, we added “Different from optimization-based schemes, backpressure optimization problems are established based on Lyapunov optimization framework with time being slotted, and the lower bounds of the time average values of optimization utilities achieved by backpressure-based schemes can be pushed arbitrarily to the optimal values[44].”

3.      In Line 68 to 71, to introduce the main content of the new references about backpressure-based schemes, “[36] combines data queues and energy queues in Lyapunov functions for cross-layer control.” is replaced by “In [41] and [42], both data queues and energy queues are utilized in Lyapunov functions for cross-layer control. In [43], the shortest path is combined with backpressure scheme in the process of next-hop node selecting to prolong network lifetime.”

Reviewer 3 Report

Authors proposed a cross-layer QoS scheme for wireless multimedia sensor networks aiming to achieve an optimal trade-off between QoS performances and network lifetime. Different techniques based on Lyapunov optimization framework have been adapted to develop an optimal joint flow control/routing/scheduling and power control scheme. The power control problem has been modelled as a convex optimization problem and optimal algorithms are used to solve the problem.  The paper is well-structured and provides reasonable amount of contribution to the filed. The algorithms are well-presented and proved accordingly.  Also, the simulation results show the performance of the proposed approach compared to existing schemes. The paper still needs to be improved in terms of grammar and language. Also, authors can improve the quality of presentation for the proposed model. 

Author Response

Dear Reviewer:

Thank you very much for giving me an opportunity to revise my manuscript, I appreciate you very much for your positive and constructive comments and suggestions on the manuscript entitled “A Cross-Layer Optimization QoS Scheme in Wireless Multimedia Sensor Networks”. (Manuscript ID: algorithms-440127). Those comments are all valuable and very helpful for revising and improving our paper, as well as the important guiding significance to my researches.

I have studied your comments carefully and have made revisions. I have tried our best to revise the manuscript according to the comments. I sincerely hope this manuscript will be finally acceptable to be published on journal Algorithms.

I would like to express my great appreciation to you for comments on the paper. Looking forward to hearing from you.

The responses to the reviewers’ comments are as following. The order of the responses follows the order of comments in the decision letters.

Point 1: The paper still needs to be improved in terms of grammar and language. Also, authors can improve the quality of presentation for the proposed model.

Response 1: Thanks for the reviewer’s kind advice.

To improve the paper in terms of grammar and language, and increase the quality of presentation for the proposed model, there are some modifications in the manuscript . The detailed modifications are as follows.

1.      In “Abstract”, Line 12, the statement “, which can” was replaced by “to”.

2.      In “Abstract”, Line 13, the statement “To adaptively allocate” was replaced by “For adaptive distribution of”.

3.      In “Abstract”, Line 15 to 16, the sentence “Normalization is used to endow different types of queues in Lyapunov function with the uniform unit.” was replaced by “In Lyapunov function, different types of queues are normalized for an unified dimension.”

4.      In “Abstract”, Line 17 to 18, the sentence “Control coefficients are designed to prolong network lifetime according to the characteristics of wireless sensor networks .” was replaced by “To prolong network lifetime control coefficients are designed according to the characteristics of wireless sensor networks.”

5.      In “Abstract”, Line20, the statement “scheme proposed” was replaced by “proposed scheme”.

6.      In “Abstract”, Line21, the statement “of” was replaced by “between”.

7.      In “Abstract”, Line22, the statement “combined with” was replaced by “utilizing”.

8.      In “Abstract”, Line22 and 23, the statement “in scheduling” was added.

9.      In “Introduction”, Line27 to 30, the sentence “Wireless multimedia sensor networks consist of non-realtime nodes which generate non-realtime data and multimedia nodes which are source nodes of audio and video sessions [1].” was replaced by “Realtime applications including multimedia applications have been developed as important services in various types of networks [1][2]. In wireless multimedia sensor networks, realtime applications and non-realtime coexist.”

10.  In “Introduction”, Line31, the statement “for” was replaced by “in”.

11.  In “Introduction”, Line31, the statement “to consider” was deleted.

12.  In “Introduction”, Line33, the statement “of” was replaced by “providing”.

13.  In “Introduction”, Line34, the statement “consumed energy” was replaced by “energy consumption”.

14.  In “Introduction”, Line36, the statement “of” was replaced by “providing”.

15.  In “Introduction”, Line37, the statement “can obtain different transmission opportunities by being” was replaced by “are”.

16.  In “Introduction”, Line38, the statement “transmission” was added.

17.  In “Introduction”, Line39, the statement “of” was replaced by “providing”.

18.  In “Introduction”, Line42, the statement “realtime” was added.

19.  In “Introduction”, Line43, the statement “insufficient” was replaced by “shortage of”.

20.  In “Introduction”, Line57, the statement “mostly” was added.

21.  In “Introduction”, Line58, the statement “in cases of nodes being” was replaced by “when nodes are”.

22.  In “Introduction”, Line59, the statement “adversely affect” was replaced by “degrade”.

23.  In “Introduction”, Line60 to 61, the statement “and constraints are designed according to network models” was deleted.

24.  In “Introduction”, Line62, the statement “optimization procedure of” was replaced by “the process of solving”.

25.  In “Introduction”, Line62 to 66, the sentence “Different from optimization-based schemes, backpressure optimization problems are established based on Lyapunov optimization framework with time being slotted, and the lower bounds of the time average values of optimization utilities achieved by backpressure-based schemes can be pushed arbitrarily to the optimal values[44].” was added.

26.  In “Introduction”, Line67 to 71, the sentence “[36] combines data queues and energy queues in Lyapunov functions for cross-layer control.” was replaced by “In [41] and [42], both data queues and energy queues are utilized in Lyapunov functions for cross-layer control. In [43], the shortest path is combined with backpressure scheme in the process of next-hop node selecting to prolong network lifetime.”

27.  In “Introduction”, Line71 to 72, the sentence “However, there are still problems ignored in existing optimization-based and backpressure-based schemes as follows.” was replaced by “However, two problems are ignored in existing optimization-based and backpressure-based schemes”.

28.  In “Introduction”, Line74, the statement “of” was replaced by “between”.

29.  In “Introduction”, Line79, the statement “of” was replaced by “between”.

30.  In “Introduction”, Line80, the statement “are also defective” was replaced by “also need improvement”.

31.  In “Introduction”, Line86 to 87, the sentence “Considering that basic units of queues are not uniform, normalization is used to set weights of queues in Lyapunov optimization framework.” was replaced by “To unify the dimension, different types of queues are normalized in Lyapunov optimization framework.”

32.  In section2.1 , Line103 to 104, the sentence “Initial energy of multimedia nodes is higher than that of common nodes.” was replaced by “Multimedia nodes are initialized with higher energy than common nodes.”

33.  In section2.1, Line104, the statement “be with” was replaced by “have”.

34.  In section2.1 , Line108 to 100, the statement “After broadcasting of local information by one node on control channel,” was replaced by “When a node broadcasts local information on control channel,”

35.  In section2.1, Line110, the statement “listening” was replaced by “monitoring”.

36.  In section2.1, Line116, the statement “just” was replaced by “either”.

37.  In section2.1, Line124, the statement “the transmission from” was added.

38.  In section2.1, Line129, the statement “In the case of” was replaced by “If”.

39.  In section2.1, Line129, the statement “being” was replaced by “is”.

40.  In section2.1, Line133, the statement “transmission” was added.

41.  In section2.1, Line134, the statement “that can be transmitted on this link” was replaced by “the link can support”.

42.  In section2.1, Line134, the statement “that can be transmitted on this link” was replaced by “the link can support”.

43.  In section2.3, Line151 to 154, the statement “is difficult to effectively support end-to-end QoS requirements of sessions according to the end-to-end situation of sessions” was replaced by “can’t adaptively distribute transmission priorities according to the end-to-end QoS requirements and the realtime end-to-end situation of sessions,”  ”

44.  In section2.3, Line157 to 159, the statement “there is one data queue maintained in the network layer of each node which contains all the data packets cached in this node.” was replaced by “each node maintains one data queue which contains all data packets at the network layer.”

45.  In section2.4, Line179 to 180, the statement “current amount of energy the node stores” was replaced by “residual energy amount”.

46.  In section2.4, Line183 to 184, the statement “the amount of energy that node n consumes to transmit data packets” was replaced by “the energy consumption for data transmission”.

47.  In section2.4, Line192 to 193, the statement “energy consumption of sensing/processing control packets of node n” was replaced by “energy consumption in control packets sensing/processing by node n”.

48.  In section2.4, Line196 to 197, the statement “that has been consumed by” was replaced by “consumption of”.

49.  In section3.1, Line201 and Line206, the statement “utility” was replaced by “utilities”.

50.  In section3.1, Line207, the statement “for” was replaced by “to”.

51.  In section3.2, Line238 to 239, the statement “Considering that basic units of queues are not uniform, normalization is used to set weights of queues” was replaced by “Normalization of different types of queues is utilized for an unified dimension”.

52.  In section3.2, Line242 to 246, the sentence “ As the sink node is the destination node of all sessions in the network, the energy of nodes around the sink node will easily be exhausted for relay of packets to the sink node.” was replaced by “In order to relay data to the sink node, energy of nodes around the sink node will be easily exhausted.”

53.  In section3.2, Line248, the statement “transmit packets with low power” was replaced by “reduce transmission power”.

54.  In section3.2, Line250, the statement “designed to decide” was replaced by “used as”.

55.  In section3.2, Line255, the statement “is in accord with” was replaced by “fits”.

56.  In section3.2, Line258, the statement “the amount of energy consumed” was replaced by “energy consumption”.

57.  In section3.2, Line262, the statement “when” was added.

58.  In section3.2, Line265 to 271, the sentence “Obviously, data queues including packets with low remaining lifetime should be transmitted first. However, as packets in  maintained using DQS belong to different services with diverse QoS requirements, as well as have different remaining lifetime in time slot t, it is not suitable to schedule according to .  is designed as the delay coefficient of  in time slot t, which is used to increase the scheduling priorities of the queues with packets having lower remaining lifetime.” was replaced by “ is the delay coefficient of in time slot t. The design principle of  is that, the lower the total remaining lifetime of packets in  is, the higher scheduling priority  has.”.

59.  In section3.2, Line274, the statement “end-to-end delay allowed” was replaced by “allowable end-to-end delay”.

60.  In section3.2, Line293, the statement “and” was added.

61.  In section3.2, Line305, the statement “a” was added.

62.  In section3.2, Line318 to 320, the statement “packets on any link are transmitted with the maximal transmission capacity of the link” was replaced by “the whole transmission capacity of any link will be devoted to data transmission on the link”.

63.  In section3.2, Line 320, the statement “far more short” was replaced by “much shorter”.

64.  In section3.2, Line 328 to 329, the statement “the set of links that can transmit data packets simultaneously” was replaced by “a link set for simultaneous transmission”.

65.  In section3.2, Line 355, the statement “by using” was replaced by “through equality”.

66.  In section3.2, Line 361, the statement “through updating  for several times” was replaced by “through multiple iterations of ”.

67.  In section3.2, Line 362, the sentence “tt denotes the update interval.” was replaced by “is the ttth iteration of .”

68.  In section3.2, Line 362, the statement “update interval” was replaced by “iteration”.

69.  In section3.2, Line 382 to 384, the sentence “Each send node updates the transmission power according to (48).” was replaced by “After gathering  from all receiving nodes, each send node updates its transmission power according to (48).”

70.  In section3.2, Line 387, the statement “several” was replaced by “multiple”.

71.  In section3.2, Line 396, the statement “equality” was added.

72.  In section3.2, Line 399 to 401, the sentence “The computational complexity increases sharply with the increase of |N|.” was replaced by “The increase of |N| will lead to a sharp rise in the computational complexity.”

73.  In section3.2, Line 405 to 406, the statement “access the medium” was replaced by “obtain transmission opportunity”.

74.  In section3.2, Line 409, the statement “defined as” was added.

75.  In section 4.1, Line 417, the statement “position information” was replaced by “location”.

76.  In section 4.1, Line 420 to 421, the sentence “The unit of transmission data amount is set as packet.” was replaced by “Data is transmitted in packets.”

77.  In section 4.1, Line 425, the statement “available” was added.

78.  In section 4.1, Line 425 to 426, the statement “can be chosen by a node” was deleted.

79.  In section 4.1, Line 437, the statement “CLADS-BCD traverses all link sets to obtains using ” was replaced by “In CLADS-BCD  are obtained through enumeration on all link sets”.

80.  In section 4.2, Line 454, the statement “where” was replaced by “requiring that”.

81.  In section 4.2, Line 458 to 468, for Fig. 1, we added the experimental analysis as “In wireless sensor networks, energy of nodes around the sink node can easily be exhausted which limits the stability periods of WSN. Under ACH, in each cluster-setup round, the nodes with more residual energy can be chosen as the cluster heads with higher probabilities, and this can balance energy consumption of nodes around the sink node. Therefore, networks under ACH have the longest stability periods. CLADS-BCD, CLADS-MAC and EASYO all consider residual energy states of nodes in scheduling to balance energy consumption. In CLADS schemes, distance coefficients for energy queues are applied to reduce energy consumption of nodes around the sink node, and this leads to longer stability periods under CLADS-BCD than under EASYO. The reason why the stability periods under CLADS-MAC are the shortest is that CLADS-MAC adopts distributed media access control scheme in scheduling with obtaining sub-optimal transmission link sets.”

82.  In section 4.2, in Line 474 to 479, for Fig. 2, we added the experimental analysis as “The reason is that, in each cluster, TDMA scheduling is adopted, which effectively reduces consumed power. Different from EASYO, in CLADS-BCD queues are normalized in Lyapunov function to unify the dimension of queues, which leads to more valid weights of energy queues in the framework and reduction of consumed power. The transmission link sets obtained by the distributed media access control scheme in scheduling of CLADS-MAC are sub-optimal, and this increases consumed power of network.”

83.  In section 4.2, in Line 483 to 485, for Fig. 3 and Fig. 4, we added the experimental analysis as “The reason is that under ACH energy consumption is well balanced using clusters. TDMA scheduling is also utilized in clusters to further reduce energy consumption.”

In section 4.2, in Line 488 to 494, for Fig. 3 and Fig. 4, we added the experimental analysis as “The reason is as follows. Transmission powers of nodes around the sink node are reduced through utilization of distance coefficients for energy queues and distance coefficients for data queues in CLADS schemes. Therefore, the average lifetime of nodes in the region of radius R under CLADS-BCD is increased a lot. CLADS also performs better than EASYO in reducing energy consumption of nodes out of the region of radius R through applying normalization of queues in optimization framework. In addition, since CLADS-BCD is optimal and CLADS-MAC is sub-optimal in scheduling, CLADS-BCD performs better than CLADS-MAC.”

84.  In section 4.2, Line 506 to 514, for Fig. 5 and Fig. 6, we added the experimental analysis as “The reason is as follows. Both the data transmission time and throughput under ACH are reduced since TDMA scheduling is utilized in clusters. Since weights of data queues in optimization framework in CLADS-BCD are lower than those in EASYO scheme, CLADS-BCD and CLADS-MAC performs worse than EASYO in average throughput. It should be noted that, average throughput of realtime sessions under CLADS-MAC is the highest. The reason is that in CLADS-MAC using distributed scheduling scheme, the nodes with more buffering packets can obtain more transmission opportunities. This means that, CLADS-MAC is suitable for wireless multimedia Ad hoc networks in which nodes are initialed with adequate energy.”

85.  In section 4.2, Line 522 to 524, for Fig. 7 and Fig. 8, we added the experimental analysis as “The reason is that in scheduling of CLADS-MAC, the nodes with more buffering packets may obtain transmission opportunities with higher probability, and this can reduce average end-to-end delay.”

86.  In section 4.2, Line 524, the statement “Considering” was replaced by “According to”.

87.  In section 4.2, Line 525, the statement “best” was replaced by “most effective”.

88.  In section 4.2, Line 531 to 536, we added the experimental analysis as “The reason is as follows. Weights of data queues in Lyapunov optimization framework as well as data transmission priorities under CLADS-BCD are lower than those under EASYO. Therefore, the average end-to-end delay under CLADS-BCD is higher. However, DQS scheme is utilized to maintain data queues in CLADS-BCD. In each data queue, transmission opportunity is distributed to the packet with the least survival time. Therefore, CLADS-BCD can provides QoS guarantees on average end-to-end delay.”

89.  In section 4.2, Line 545 to 549, for Fig. 9, we added the experimental analysis as “The reason is as follows. According to the characteristics of wireless sensor networks, CLADS-BCD adopts distance coefficients of energy queues in optimization to reduce energy consumption of nodes in the surrounding area of the sink node. This can prolong lifetime of nodes near the sink node, as well as increase network lifetime.”

90.  In section 4.2, Line 554, the statement “the sink node in the” was deleted.

91.  In section 4.2, Line 554 to 556, for Fig. 10 and Fig. 11, we added the experimental analysis as “The reason is that the network lifetime of CLADS-BCD is much higher than that of EASYO. Then, the sink node can collect data packets for much more time under CLADS-BCD than under EASYO.”

92.  In section 4.2, Line 566, the statement “transmission of realtime services” was replaced by “realtime applications”.

93.  In section 4.2, Line 566, the statement “whose nodes are with adequate energy” was replaced by “with nodes having adequate energy”.

94.  In section 4.2, Line 570, the statement “of” was replaced by “between”.

95.  In section 4.2, Line 572, the statement “of” was replaced by “between”.

96.  In section 4.2, Line 576 to 580, the sentence “The distributed media access control scheme can be used to reduce computational complexity of scheduling, as well as be used for transmission of realtime services in wireless Ad hoc networks.” was replaced by “The proposed cross-layer scheme combined with low computational complexity distributed media access control scheme is suitable for realtime services in wireless Ad hoc networks.”

Round  2

Reviewer 1 Report

Comments have been answered appropriately.

Reviewer 2 Report

The reviewers' comments have been addressed.

Reviewer 3 Report

Authors have covered all my comments and improved the quality of the paper. I agree to accept the paper in present form.